# *Clostridium perfringens* Epsilon Toxin Binds to and Kills Primary Human Lymphocytes

**DOI:** 10.3390/toxins15070423

**Published:** 2023-06-29

**Authors:** Samantha V. Shetty, Michael R. Mazzucco, Paige Winokur, Sylvia V. Haigh, Kareem Rashid Rumah, Vincent A. Fischetti, Timothy Vartanian, Jennifer R. Linden

**Affiliations:** 1Feil Family Brain and Mind Research Institute, Weill Cornell Medical College, New York, NY 10065, USAtiv2002@med.cornell.edu (T.V.); 2Harold and Margaret Milliken Hatch Laboratory of Neuro-Endocrinology Rockefeller University, New York, NY 10065, USA; 3Laboratory of Bacterial Pathogenesis and Immunology, Rockefeller University, New York, NY 10065, USA

**Keywords:** epsilon toxin, ETX, myelin and lymphocyte protein, MAL, clostridium perfringens, lymphocytes, CD4+ cells, CD8+ cells, CD19+ cells, B cells, T cells

## Abstract

*Clostridium perfringens* epsilon toxin (ETX) is the third most lethal bacterial toxin and has been suggested to be an environmental trigger of multiple sclerosis, an immune-mediated disease of the human central nervous system. However, ETX cytotoxicity on primary human cells has not been investigated. In this article, we demonstrate that ETX preferentially binds to and kills human lymphocytes expressing increased levels of the myelin and lymphocyte protein MAL. Using flow cytometry, ETX binding was determined to be time and dose dependent and was highest for CD4+ cells, followed by CD8+ and then CD19+ cells. Similar results were seen with ETX-induced cytotoxicity. To determine if ETX preference for CD4+ cells was related to MAL expression, MAL gene expression was determined by RT-qPCR. CD4+ cells had the highest amount of Mal gene expression followed by CD8+ and CD19+ cells. These data indicate that primary human cells are susceptible to ETX and support the hypothesis that MAL is a main receptor for ETX. Interestingly, ETX bindings to human lymphocytes suggest that ETX may influence immune response in multiple sclerosis.

## 1. Introduction

Epsilon toxin (ETX) is produced by *Clostridium perfringens* (*C. perfringens*) toxinotypes B and D and is the third most lethal bacterial toxin known to date. Recently, ETX has been identified as a possible environmental cause of multiple sclerosis (MS) in humans, an immune-mediated, demyelinating disease of the central nervous system (CNS) [1,2,3]. Our group has demonstrated that MS patients are more likely to be colonized by ETX-producing *C. perfringens* toxinotypes versus healthy controls (HC), 61% versus 13%, respectively [3]. In addition, in colonized individuals, the relative abundance of ETX producing strains is significantly higher in MS patients versus HC. Histopathological, active MS lesions are characterized by overt blood–brain-barrier permeability, demyelination, and robust CNS immune infiltration. Using both in vitro and in vivo models, we have demonstrated that ETX specifically causes BBB permeability [4,5], demyelination [6], and loss of CNS-immune privilege [3]. Remarkable, this specificity is mediated by the selective expression of the ETX receptor, the myelin and lymphocyte protein MAL [7], on CNS endothelial cells [5], mature oligodendrocytes [6], and human lymphocytes [8,9] (murine lymphocytes do not express MAL [10]). Importantly, pathogenic lymphocytes, including both T cells and B cells, play an important role in MS pathogenesis [11,12]. However, whether ETX targets primary human lymphocytes or plays a role in MS pathogenesis is currently unknown. In this paper, we sought to determine if ETX can bind to and kill primary human lymphocytes expressing MAL.

ETX is secreted as a relatively inactive 33 kDa protoxin (pETX), and proteolytic cleavage at the N and C terminal by trypsin and chymotrypsin (or bacterial lambda toxin) results in 1000 fold increase in ETX’s toxicity [13]. Cytotoxicity is mediated by pore formation/oligomerization, resulting in ionic imbalances [14,15,16]. In ruminant livestock, ETX is responsible for causing an economically devastating disease, enterotoxaemia [17,18]. In both natural and experimental infections, ETX preferentially accumulates in the brains and kidneys of animals [19,20,21,22,23,24,25]. In the brain, ETX selectively binds to brain endothelial cells, myelinated structures, and mature oligodendrocytes, the myelinating cells of the CNS [5,6,26,27,28,29,30]. In the kidney, ETX preferentially binds to distal and collecting tubules [22,24]. ETX preferences for these specific cell types are most likely due to the expression of the ETX receptor MAL.

Although a handful of proteins have been identified as possible receptors of co-receptors for ETX [7,31,32], MAL expression is the only protein to have been demonstrated to be both necessary and sufficient for ETX binding and cell death. Importantly, this protein has been shown to be necessary for ETX binding and cytotoxicity in both gain-of-function and loss-of-function assays and has been demonstrated to directly interact with ETX via immunoprecipitation assays. Expression of both human and rodent MAL in the normally ETX-resistant Chinese hamster (CHO) line conferred both ETX binding and cytotoxicity [7]. In addition, MAL deficiency (knockout) abolished ETX binding to tissue [5,7]. Recently, these results have been confirmed in an independent study that again demonstrated that (1) the expression of endogenous human MAL in ETX-resistant cell lines confirmed ETX binding and cytotoxicity, and (2) the deletion of MAL expression in normally ETX-sensitive cell lines abolished ETX binding and cytotoxicity [9]. Importantly, these authors were able to demonstrate the direct protein–protein interaction of MAL and ETX via immunoprecipitation. Finally, MAL expression was also necessary for ETX-induced oligodendrocyte cell death and demyelination as well as the ETX-induced blood–brain barrier permeability [5,6]. Based on these various loss-of-function and gain-of-function assays, we believe that MAL is the strongest candidate as the main ETX receptor.

MAL is a proteolipid, tetraspan protein with two extracellular loops. MAL cDNA was first cloned from human T cell lines and is indicated to play an important role in T cell differentiation, membrane protein sorting, and formation of immunological synapses [8,33,34,35,36,37,38]. Since then, MAL has also been cloned from oligodendrocytes, myelin, MDCK cells, whole mouse brains, and found to be expressed in certain polarized epithelial cells [8,39,40,41,42,43]. MAL localizes to lipid rafts and plays an important role in lipid raft formation, stabilization, and maintenance [37,44,45,46]. In addition, MAL plays a significant role in protein cycling from the trans-golgi network to the plasma membrane and is important for apical protein sorting in polarized cells [42,46,47,48,49,50,51,52,53]. In the nervous system, MAL is believed to play an important role in myelin biogenesis and proper axon–glial interactions [41,54].

In human lymphocytes, MAL expression has been reported to be confined to the T cell lineage; expression of MAL in healthy B cell lineages is less clear [8,33,34,35,37,38,55,56,57,58,59]. Recently, it has been demonstrated that ETX can bind to and kill cancerous cell lines of the T cell lineage which express MAL but not cell lines of the B cell lineage which do not express MAL [9]. However, as MAL expression has been reported to be dysregulated in cancer, including cancerous lymphocytic cell lines [56,60,61,62,63,64,65], it is imperative that ETX-lymphocyte interactions be confirmed in primary, human lymphocytes isolated from healthy donors. The aim of this study was to determine if ETX could bind to and kill primary human lymphocytes that endogenously express MAL, specifically, human CD4+, CD8+, and CD19+ lymphocytes. We hypothesize that lymphocytes with higher amounts of MAL expression will have a higher affinity for and thus an increased susceptibility to ETX.

## 2. Results

### 2.1. Primary Human Lymphocytes Express Mal

To confirm *Mal* gene expression in the T cell lineage, real-time quantitative PCR (RT-qPCR) for human *Mal* was performed on isolated CD4+, CD8+, and B cells (Figure 1). *Mal* gene expression was normalized to CD4+ cells. RT-qPCR analysis confirmed that CD4+ cells had the highest amount of *Mal* gene expression compared to isolated CD8+ and B cells (Figure 1). In addition, CD8+ cells showed a trend towards expressing significantly more *Mal* than B cells.

The detection of *Mal* expression in B cells was unexpected and may be due to T cell contamination during our isolation method. To determine if the low Mal expression observed in our isolated B cells populations were a result of T cell contamination, we compared our *Mal* expression results to those of other publicly available datasets using a variety of cell isolation and gene expression techniques (Appendix A) [66,67,68,69]. Isolation methods included FACS sorting (Appendix A), positive magnetic selection (Appendix A), and single-cell RNAseq analysis (Appendix A), whereas *Mal* expression was evaluated using RNAseq (Appendix A) and a microarray (Appendix A). The examination of these four independent datasets confirmed significantly higher *Mal* gene expression in CD4+ cells, followed by CD8+ cells, and, finally, CD19+/B cells. These results also demonstrated a low but still detectable level of *Mal* transcripts in CD19/B cells, consistent with our RT-qPCR results. Based on these findings, we believe our RT-qPCR results are accurate.

### 2.2. ETX Binds to Human Lymphocytes with a Preference for CD4+ Cells

To determine if the ETX bound to human lymphocytes expressing *Mal,* PBMNCs were probed with 50 nM of Alexa Fluor 647 pETX (pETX-647) for 2 h, and binding to CD4+, CD8+, and CD19+ cells evaluated by multicolor flow cytometry (Figure 2A–C). pETX was used to study ETX binding because pETX binds with a similar affinity as active ETX but does not oligomerize and form pores, preventing endosome recycling and the possible cell surface rearrangement of MAL [5,7,15,34,37,38,70,71,72,73,74,75,76,77]. Untreated cells (0 nM) were used as negative controls. Scatter plots (Figure 2A) and histogram analyses of pETX-647 fluorescent intensities (Figure 2B) revealed that CD4+ cells bound more toxins than CD8+ and CD19+ cells. In addition, CD8+ cells bound more toxins than CD19+ cells.

To quantify ETX bindings to target cells, pETX-647 binding was quantified via flow cytometry after 2 h of incubation with 25 nM of pETX-647. Significantly more CD4+ cells were positive for pETX-647 compared to CD8+ and CD19+ cells; 82.3%, 60.3%, and 24.7%, respectively (Figure 2C). Even when cells were incubated with 25 nM of pETX-647 for 15 min, 50.1%, 30.0%, and 18.2% of CD4+, CD8+, and CD19+ cells were positive for pETX-647, respectively. This trend was observed for all investigated time points.

Remarkably, pETX-647 could be observed binding to CD4+ cells at concentrations as low as 1 nM (Figure 2D). PBMNCs were incubated with 1 nM of pETX-647 for 2 h, and percent positive cells were evaluated by flow cytometry. Untreated cells (0 nM) were used as negative controls. No significant differences were observed in CD8+ or CD19+ when cells were treated with or without 1 nM pETX-647. In comparison, significantly more CD4+ cells were positive for pETX-647 when treated with 1 nM pETX-647 than without, 0.34% versus 0.07%, respectively.

To confirm that pETX bindings to lymphocytes is pETX-specific, pETX-647 was pretreated with an anti-ETX antibody shown to block ETX binding [78]. Binding to total lymphocytes was inhibited when media containing pETX-647 was pre-treated with the anti-ETX antibody (Figure 2E). To ensure that the fluorescent signal observed in lymphocytes was not due to excess fluorophore from the pETX-647 labeling process, PBMNCs were treated with shiga toxin (STX) fluorescently conjugated with Alexa Fluor 647 using the exact same labeling process (STX-647) (Appendix A). When PBMNCs were incubated with 50 nM of STX-647 for 2 h, only a small percentage of lymphocytes bound STX-647, less than 2%. Importantly, the percentage of CD19+ cells positive for STX-647 was significantly higher than CD4+ or CD8+ cells: 1.64%, 0.19%, and 0.01%, respectively, confirming previous results that B cells had increased affinity for STX [79]. The low percentage of STX-647-positive cells indicates that the contamination of lymphocytes by excess dye when treated with Alexa Fluor conjugated 647 toxin is minimal.

Finally, we sought to confirm that probing cells with pETX was a reliable marker for ETX binding. PBMNCs were probed with 25 nM ETX or pETX for 2 h, and bindings were determined using an affinity purified anti-ETX polyclonal rabbit antibody and PE-conjugated anti-rabbit IgG and examined by flow cytometry (Figure 2F). Cells treated without ETX were used as controls. No significant differences in ETX and pETX binding were observed. Similar results were obtained when cells were probed with 5 nM, 10 nM, and 50 nM of ETX or pETX as well (data not shown). This confirms the previously published results that pETX and ETX bound similarly to target cells [4,24].

Taken together, this data indicates that ETX specifically binds to human primary lymphocytes, with a preference for CD4+ cells, followed by CD8+ and then CD19+ cells. Importantly, ETX binding positively associates with *Mal* gene expression.

### 2.3. ETX Bindings to Human Lymphocytes Is Dose and Time Dependent

To determine if the ETX bindings to lymphocyte subsets were dose and time dependent, PBMNCs were incubated with 0 nM, 1 nM, 5 nM, 10 nM, 25 nM, and 50 nM of pETX-647 for 15, 30, 60, and 120 min. After 15 min of incubation, pETX was observed binding to CD4+, CD8+, and CD19+ cells in a dose-dependent manner (Figure 3A). For a full breakdown of *p* values between different doses, please refer to Appendix A. After 15 min, significantly more CD4+ cells were positive for ETX when treated with 10 nM (11%), 25 nM (50%), and 50 nM (75%) compared to untreated controls (0%) (Figure 3A). In comparison, a significant increase in ETX-positive CD8+ and CD19+ cells was not observed until treatment with 25 nM pETX-647. When PBMNCs were incubated for 120 min with pETX-647, significantly more CD4+ cells were positive for ETX at 5 nM (41%), 10 nM (62%), 25 nM (83%), and 50 nM (90%) compared to untreated controls (0%) (Figure 3B). In contrast, a significant increase in pETX-647 positive CD8+ and CD19+ cells was not observed until cells were treated with 10 nM pETX-647. A similar trend was observed when cells were incubated with pETX-647 for 30 and 60 min (Appendix A, respectively). These data indicated that the ETX bindings to all lymphocyte subsets is dose dependent and reaffirms that ETX preferentially binds to CD4+ compared to CD8+ and CD19+ cells.

For CD4+ cells, significant differences in ETX bindings at different time points were observed when cells were treated with 1 nM, 5 nM, 10 nM, or 25 nM pETX-647 (Figure 3C). Remarkably, after 1 nM treatment for 15 min, 0.021% of CD4+ cells were positive for ETX. After 60 and 120 min, significantly more CD4+ cells were positive for pETX: 0.060% and 0.067%, respectively. In comparison, with 5 nM treatment, 3.2%, 8.1%, 23.8%, and 40.9% of CD4+ cells were positive for pETX after 15-, 30-, 60-, and 120-min incubations, respectively. Similar trends were seen for 10 nM and 25 nM doses. At 50 nM pETX-647 treatment, pETX binding to CD4+ cells appeared to be saturated, as there were no significant differences between any of the time points. Similar results were observed in CD8+ cells (Figure 3D), with the clearest time-dependent binding occurring with 10 nM pETX-647 treatment. In total, 4.4%, 17.2%, 21.6%, and 34.2% of CD8+ cells were positive for pETX after 15-, 30-, 60-, and 120-min incubations, respectively. Again, pETX binding appeared to be saturated for all time points at 50 nM treatment for CD8+ cells. Although ETX binding was observed on CD19+ cells, binding did not appear to be time dependent under these conditions (Figure 3E). These data indicated that the ETX bindings to CD4+ and CD8+ cells were time dependent and, again, reaffirmed that ETX preferentially bound to CD4+ cells, compared to CD8+ and CD19+ cells.

### 2.4. ETX Induces Cytotoxicity in Human Lymphocytes, Especially CD4+ Cells

To determine if ETX binding to human lymphocytes conferred cytotoxicity, total lymphocytes were evaluated for cell death by propidium iodide (PI) inclusion via flow cytometry (Figure 4A,B). Cells positive for (PI+) were considered dead. After 4 h of active ETX treatment, only a small percentage of cell death was observed: 1.4% (Figure 4C). A significant increase in total lymphocyte cell death was observed at ETX doses of 25 nM and 50 nM: 11.5% and 17.8%, respectively. Importantly, pretreatment of ETX with a neutralizing antibody that blocks ETX-cytotoxicity inhibited ETX-induced cell death (Figure 4D) [78].

To determine if lymphocyte populations expressing higher levels of *Mal* were more susceptible to ETX-induced cytotoxicity, cell death was evaluated in CD4+, CD8+ and CD19+ cells by flow cytometry (Figure 4E). When treated with 25 nM of ETX for 4 h, cell death was significantly higher in CD4+ cells compared to CD8+ and CD19+ cells: 19.6%, 6.5%, and 3.0%, respectively. Similar results were seen when cells were treated with 50 nM ETX with CD4+, CD8+, and CD19+ cells, exhibiting 35.1%, 13.9%, and 1.9% cell death, respectively. In addition, CD8+ cell death was significantly higher than CD19+ cell death. Taken together, these data demonstrate that ETX induced cell death in CD4+ and CD8+ cells is positively associated with *Mal* gene expression.

### 2.5. ETX-Induced Cytotoxicity in Human CD4+ Cells Is Time and Dose Dependent

To determine if ETX-induced cytotoxicity in CD4+ cells is dose dependent, PBMNCs were incubated with 0 nM, 1 nM, 5 nM, 10 nM, 25 nM, and 50 nM of active ETX for 4 h (Figure 4F). Significant increases in percent cell death compared to untreated controls (0.82%) were observed at 25 nM (19.6%) and 50 nM (35.1%) doses. Cell death at 50 nM was significantly higher than at 25 nM treatment. This indicates that ETX-induced cell death of CD4+ cells is dose dependent.

To determine if ETX-induced cell death of CD4+ cells is time-dependent, PBMNCs were incubated with indicated doses of ETX for 30, 60, 120, and 240 min (Figure 4G). At a dose as low as 1 nM, a significant increase in CD4+ cell death was observed between 30 min and 4 h: 0.55% and 1%, respectively. At a dose of 5 nM, a significant increase in CD4+ cell death was observed between 30 min and 4 h: 0.47% and 1.21%, respectively. Cell death was considerably higher at larger ETX doses. At 50 nM ETX treatment, cell death was 0.84%, 2.06%, 18.06%, and 35.07% after 30 min, 60 min, 2 h, and 4 h of treatment, respectively. This data indicates that ETX induced CD4+ cell death in a time-dependent.

### 2.6. ETX-Induced Cytotoxicity in Human Lymphocytes Is Mediated by Pore Formation

ETX-induced cell death has been proposed to be mediated by pore formation in sensitive cell lines. To determine if ETX pore formation occurs in primary human lymphocytes, whole-cell lysates from PBMNC treated with 0 nM, 10 nM, 25 nM, or 50 nM ETX for two hours were evaluated by Western blot analysis (Figure 5A). Lysates from control and ETX-treated rMAL-CHO cells, known to form a 150 kDA ETX pore complex [78], were used as positive controls. The 150 kDA pore complex was only observed when PBMNCs were treated with 50 nM ETX (Figure 5A). In addition, a band at 27 kDA could be observed in all cell lysates treated with 50 and 25 nM ETX, indicating bound ETX monomers, as all cells were thoroughly washed in PBS prior to lysis. Pore formation appeared to be time dependent when cells were treated with 50 nM of ETX for 30, 60, and 120 min (Figure 5B). This indicates that ETX-mediated cytotoxicity of PBMNCs is mediated by pore formation.

## 3. Discussion

### 3.1. ETX Binding and Cytotoxicity Positively Associates with Mal Gene Expression in Human Lymphocytes

In this paper, we demonstrate that ETX binding and cytotoxicity to primary human lymphocyte populations positively associates with *Mal* gene expression. Specifically, CD4+ cells have the highest amount of *Mal* gene expression, followed by CD8+ and then B cells. Accordingly, ETX preferentially binds to and kills CD4+ cells, followed by CD8+ and then CD19+ cells. This builds on previously published data demonstrating that ETX can bind to and kill human cell lines of the cancerous T cell lineage and is MAL dependent [9].

We first confirmed *Mal* expression in CD4+, CD8+, and B cells using RT-qPCR. The results demonstrated that CD4+ had the highest amount of *Mal* gene expression, followed by CD8+ cells and then CD19+ cells. Increased *Mal* gene expression in CD4+ cells was also confirmed using publicly available datasets [66,67,68,69]. Secondly, we demonstrated that ETX bound to human lymphocytes with a preference that positively associates with *Mal* expression. When lymphocytes were incubated with 25 nM of pETX-647 for two hours, 82.8% of CD4+, 60.3% of CD8+, and 24.7% of CD19+ cells were bound to pETX. In addition, pETX binding occurred in a dose- and time-dependent manner. Finally, we demonstrated that ETX induced lymphocyte cell death. When lymphocytes were incubated with 50 nM of active ETX for four hours, 35.1% of CD4+ and 13.9% of CD8+ exhibited cell death; no significant amount of cell death was observed in CD19+ cells. ETX-induced cell death of CD4+ cells was also dose and time dependent. Taken together, this data indicates that ETX binding and cytotoxicity to human lymphocytes is positively associated with *Mal* expression. However, these results were limited, as they only analyzed *Mal* gene expression and not protein expression. Future experiments need to confirm that *Mal* gene expression is reflective of MAL protein expression, as we were unable to complete protein analysis experiments at this time. Alternatively, ETX’s increased binding to and activity on CD4+ cells may have also been a result of CD4 expression itself. In a single experiment, ETX was observed to bind to recombinant human CD4 immobilized on Dynabeads [32], raising the possibility that CD4+ affinity and sensitivity to ETX may be a combination of both CD4 and MAL expression. Importantly, other ETX-sensitive cell lines, including MDCK and ACHN cells, also express MAL [42,47,80]. Taken together, these observations supported the theory that MAL is the main receptor for ETX.

### 3.2. MAL Expression in Human Lymphocytes

MAL expression in CD4+ and CD8+ cells was consistent with previously published results, looking at both peripheral blood lymphocytes and various cell lines of the T- and B-cell lineage. By using a privately generated anti-MAL antibody [81], Copie-Bergman et al. demonstrated that 65–90% of CD4+ and 22–39% of CD8+ cells were positive for MAL via flow cytometry [56]. In comparison, the authors did not detect a significant amount of MAL expression on B cells from peripheral blood, tonsils, or spleens: 0–0.6%, 1.5–2%, and 0.6–0.7%, respectively. They did, however, observe the occasional MAL-positive plasma cell via immunohistochemistry in tonsils or reactive lymph nodes. In addition, other groups have observed MAL expression in various T cell lines but not B cell lineages [8,33,35,37,57,58]. Conflicting results for MAL detection in B cells may have been a result of technical differences in experimental approaches and sensitivity (for example, protein expression versus gene expression). However, results consistently indicated that B cells expressed significantly lower to no MAL compared to T cells.

### 3.3. MAL’s Function in Different Lymphocyte Populations

The reason for the differential expression of MAL in specific lymphocyte populations is unknown. MAL’s function in lymphocytes has only been extensively studied in the T cell lineage. In general, MAL appears to play an important role in lipid raft formation and stabilization [36,37,44,45,46] and protein trafficking to the apical plasma membrane in polarized cells [42,46,48,49,51,52]. In human T cells, MAL is selectively present in glycolipid-enriched membrane microdomains (also known as detergent-resistant membranes) and appears to play an important role in T cell activation, mainly through its interactions with Lck, a src-like kinase [33,34,35,55].

Src-like kinases, especially Lck, play an essential role in T cell activation and maturation [82]. Previous studies have shown that MAL and Lck co-immunoprecipitate with each other in a lipid-dependent interaction in T cells [35]. If the expression of MAL is lost, Lck targeting to the plasma membrane becomes dysfunctional. As such, the loss of MAL results in the defective polarization of the T cell receptor for antigen (TCR) and the organization of the immunological synapse (IS) [33]. MAL targets Lck to the plasma membrane via vesicle movement along microtubule tracks and requires participation of Inverted Formin2 (INF2) as well as Cdc42 and Rac1 [55]. MAL has also been shown to be necessary for proper receptor and signaling protein assembly at the IS in the supramolecular activation cluster (SMAC). The incorrect localization of MAL results in Lck being transported to the wrong part of the SMAC [34]. In addition, more recent publications have also demonstrated that MAL plays an important role in endosome trafficking and exosome secretion from T cells [38].

Although MAL’s function in B cells is unknown, it is possible that MAL could play a similar role in lipid raft-protein organization and signaling in B cells. Lipid rafts play a role in B cell activation and can act as platforms for B cell receptor (BCR) signaling and possibly antigen trafficking [83]. MAL may play a similar role in Lck or other src-like kinase trafficking in B cells. Interestingly, MAL is highly expressed in mediastinal large B cell lymphoma [59,64,84] and a subset of Hodgkin lymphoma with poor prognosis [65].

### 3.4. ETX-Induced Cell Death Pathways

It is generally accepted that ETX causes cell death via the formation/oligomerization of a heptameric pore. ETX pore formation occurs in three sequential steps: (1) ETX binding to its receptor, (2) oligomerization of the pre-pore complex on the cell surface, and (3) the pore insertion into the cell membrane [15]. Pore formation results in a rapid decrease in transmembrane resistance and the rapid depletion of intracellular K^+^ and Cl^−^ [85,86]. This is followed by a slower intracellular increase in Na^+^ and Ca^2+^ [85,87]. ETX also causes a rapid depletion of ATP and causes mitochondrial membrane permeabilization and translocation of an apoptotic-inducing factor to the nucleus [86].

In our current experiments, the ETX treatment of PBMNCs resulted in ETX oligomerization/pore formation, as detected by Western blot. However, the majority of the ETX detected in the PBMNC lysates was observed as bound monomers, not in the pore complex. It is interesting to note that 90% of CD4+ cells were positive when PBMNCs were probed with 50 nM of pETX-647 for 2 h. However, when cells were treated with 50 nM of active ETX for 4 h, only 35% of the CD4+ cells died, indicating a large discrepancy in ETX binding versus cell death in CD4+ cells for this dose and time point. Alternatively, in rMAL-CHO cells, a highly ETX-susceptible cell line, we saw a closer correlation of ETX binding and ETX cytotoxicity [7,78]. When rMAL-CHO cells were treated with ETX, the vast majority of cells bound to ETX and also died. For example, when treated with 50 nM of ETX, cell viability decreased to almost 0%. In addition, pore formation happened rapidly (within 5 min, when cells were treated with 50 nM ETX) and at very low doses (within 30 min after cells were treated with 1 nM ETX), with the majority of ETX detected in the poor complex, not as bound monomers [78]. This indicates that the low amount of cell death observed in CD4+ cells despite the high binding percentage may be due to the low amount of ETX oligomerization/pore formation observed in these cells.

### 3.5. Possible Role of ETX-Lymphocytes Interactions in MS Pathogenesis

Based on the limited amount of ETX-induced cell-death observed in human lymphocytes, despite a significantly higher degree of ETX binding, we postulate that ETX binding to lymphocytes may influence other cellular behaviors in addition to cell death, including various immune functions. This raises the interesting hypothesis that ETX bindings to MAL on human lymphocytes at sublethal doses may modify immune function, possibly influencing lymphocyte activity in immune-mediated diseases such as MS. CD4+, CD8+, and B cells have all been implicated in MS pathogenesis, although the exact mechanisms by which they influence MS pathogenesis is still unclear [11,12,88,89,90,91,92,93]. It is believed that pathogenic lymphocytes, including autoreactive and proinflammatory lymphocytes, are stimulated in the periphery, prior to the infiltration of these cells into the CNS. Histopathological examination of active MS lesions reveals dense lymphocytic infiltration into the CNS perivascular space with more limited extravasation into the CNS parenchyma. These infiltrates are heavily dominated by the presence of CD4+ and CD8+ with a much lower presence of B cells. Based on these observations, MS pathology has historically been viewed as being T cell driven; however, the wide success of B cell depleting therapies in treating MS has highlighted the importance of B cells in MS pathogenesis as well. Due to MAL’s role in T cell activation and the important role it plays at the immunological synapse [33,34,35,55], it seems possible that ETX bindings to MAL may initiate a wide array of signaling cascades. Indeed, other pore-forming toxins have been shown to induce numerous cell signaling cascades not related to membrane permeabilization [94,95,96,97,98,99,100,101,102,103]. However, more examination into ETX’s impact on lymphocyte function is needed and is an area of ongoing research.

### 3.6. Conclusions

In conclusion, we have demonstrated that ETX binding and cytotoxicity to human lymphocytes positively associates with MAL gene expression, supporting the hypothesis that MAL is the main receptor for ETX.

## 4. Materials and Methods

### 4.1. Peripheral Blood Isolation from Healthy Controls

Peripheral blood samples were collected from healthy controls via the cubital vein using BD Vacutainer K2 EDTA 7.2 mg Blood Collection tubes in accordance with our Institutional Review Board, protocol number 1003010940. At the time of donation, healthy controls were free of any chronic or acute disease, were both male and female, ranged in age from 18 to 59 years old, and lived in the New York City metropolitan area.

### 4.2. Isolation of CD4+, CD8+, and B Cells from Human Peripheral Blood for RT-qPCR Analysis

Peripheral blood samples were collected from healthy controls via the cubital vein using BD Vacutainer K2 EDTA 7.2 mg Blood Collection tubes. Subsets were isolated using RossetteSep immunodensity negative selection cocktails (Stem Cell Technologies, Vancouver, BC, Canada). CD4+, CD8+, and B cells were isolated using RosetteSep™ Human CD4+ T Cell Enrichment Cocktail, RosetteSep™ Human CD8+ T Cell Enrichment Cocktail, and RosetteSep™ Human B Cell Enrichment Cocktail, respectively, per manufacturer’s instructions.

### 4.3. Real-Time Quantitative PCR (RT-qPCR) Analysis

Comparative RT-qPCR was performed in an ABI Taqman 7900HT Fast Real-Time PCR machine (Applied Biosystems, Waltham, MA, USA), using the PowerUp^TM^ SYBR^TM^ Green Master Mix (Applied Biosystems, Waltham, MA, USA). Briefly, PCR was carried out in a 10 μL volume in a final concentration of 1X SYBR^TM^ Green Master Mix containing 300 nM forward and reverse primers and 10 ng cDNA. The primer sequences were as follows: human MAL, F 5′-GGGTGATGTTCGTGTCTGTG-3′, R 5′-ACTGAGGCGCTGAGGTAAAA-3′; human b-actin, F 5′-CACCAACTGGGACGACAT-3′, R 5′-ACAGCCTGGATAGCAACG-3′. The PCR reaction steps were as follows: 50 °C for 2 min, 95 °C for 2 min, and 40 cycles of 95 °C for 15 s followed by 60 °C for 1 min. A subsequent dissociation curve measurement from 60 °C to 95 °C was carried out. All samples were run in triplicate. PCR data were analyzed using the 7900 SDS v2.4.1 software (Applied Biosystems, Waltham, MA, USA). Relative gene expression was quantified by performing double delta Ct analysis (2^−ΔΔCt^). B-actin Ct values were used as internal controls, and the gene of interest (GOI) expression was normalized to CD4+ cell expression.

### 4.4. Preparation of Fluorescently Labeled pETX

pETX was provided by BEI (at a minimum >95% purity at 0.5 mg/mL (Epsilon Protoxin, from Clostridium perfringens, Strain 34 Type B, NR-856). pETX was labeled with Alexa Fluor 647 Protein Labeling Kit (Life Technologies, Carlsbad, CA, USA) per manufacturer’s instructions. Labeled toxin was stored in a 50% glycerol stock (10 uM) at −20 °C until use.

### 4.5. Activation of ETX

pETX provided by BEI was activated in house using immobilized trypsin, TPCK Treated, agarose resin (Thermo Fischer Scientific, Waltham, MA, USA). Briefly, 125 μL resin was washed three times in sodium phosphate buffer (pH 7.98). Resin was suspended in 200 μL sodium phosphate buffer and combined with 500 uL of BEI pETX (0.5 mg/mL) for two hours at 37 °C with gentle agitation. The solution was centrifuged at 18,000 rcf for 10 min, and the supernatant containing the activated ETX was collected. ETX activation was confirmed by the treatment of rMAL-CHO cells [7] to in-house controls. Activated toxin (~11 μM) was aliquoted and stored at −80 °C until use.

### 4.6. PBMNC Isolation from Human Peripheral Blood for ETX Binding and Cytotoxicity Studies

Blood samples were collected from healthy controls via the cubital vein using BD Vacutainer K2 EDTA 7.2 mg Blood Collection tubes. Samples reached room temperature and were then diluted with an equal volume of Phosphate Buffered Saline (PBS) +2% Fetal Bovine Serum (FBS). Diluted blood was layered on top of Ficoll-Paque PLUS (GE Healthcare Bio-Sciences, Uppsala, Sweden). Tubes were centrifuged at 1200 rcf for 20 min at room temperature (without brakes). Buffy coat containing peripheral blood mononuclear cells (PBMNCs) was collected using a sterile transfer pipet. Buffy coat was washed with 20 mL of PBS + 2% FBS and centrifuged at 250 rcf for 10 min at 4 °C. The supernatant containing platelets was removed by aspiration. Cell pellet was washed with 20 mL of PBS + 2% FBS and centrifuged at 500 rcf for 10 min at 4 °C. The supernatant was aspirated, and cells were re-suspended in CTS™ OpTmizer™ T Cell Expansion Media (Thermo Fischer Scientific, Waltham, MA, USA) supplemented with Glutamax and 5% FBS. Cells were enumerated using a hemocytometer and adjusted to 1.5 × 10^6^ cells/mL. Cells were kept on ice until use.

### 4.7. Evaluation of pETX-647 Binding to Lymphocyte Subsets

To determine ETX binding, PBMNC (1.5 × 10^6^ cells/mL) were incubated with pETX-647 at 0 nM, 1 nM, 5 nM, 10 nM, 25 nM, and 50 nM for 15, 30, 60, and 120 min at 37 °C. In select experiments, media containing 50 nM pETX-647 was pretreated with or without an anti-ETX antibody (JL004) [78] for 30 min before treating cells for 2 h. At selected time points, 100 uL of cells were transferred to round bottom plates containing PBS + 2% FBS and immediately washed with PBS to remove unbound pETX-647. Cells were centrifuged at 500 rcf for 5 min. Cells were resuspended in Cell Staining buffer (Biolegend, San Diego, CA, USA) containing 5% Human TruStain FcX™ Fc Receptor Blocking Solution (BD Bioscience) for 10 min. Cells were then probed with FITC conjugated anti-CD4 Multiclone SK3 and SK4 (Biolegend, San Diego, CA, USA), PE-conjugated anti-CD8β Clone 2ST8.5H7 (BD Bioscience, San Diego, CA, USA), and V450 conjugated anti-CD19 clone SJ25C1 (BD Bioscience, San Jose, CA, USA) for 20 min at room temperature. Cells were washed and resuspended in PBS and analyzed using a BD FACSVerse Flow Cytometer (BD Bioscience, San Jose, CA, USA). From pETX-647 treatment to analysis via flow cytometry, cells were washed a total of three times. Data were collected using FACSuite™software (BD Bioscience, San Jose, CA, USA) and analyzed using FlowJo software (BD Bioscience, San Jose, CA, USA).

### 4.8. Evaluation of ETX-Induced Cytotoxicity in Lymphocyte Subsets

To determine ETX-induced cytotoxicity, PBMNC (1.5 × 10^6^ cells/mL) were treated with activated ETX at 0 nM, 1 nM, 5 nM, 10 nM, 25 nM, and 50 nM for 30, 60, 120 and 240 min at 37 °C. In select experiments, media containing 50 nM ETX was pretreated with or without an anti-ETX antibody (JL008) [78] for 30 min before treating cells for 2 h. At selected time points, 100 uL of cells were transferred to round bottom plates containing ice cold PBS + 2%FBS to stop ETX activity [15]. Cells were immediately washed to remove unbound ETX and centrifuged at 500 rcf for 5 min at 4 °C. Cells were resuspended in Cell Staining buffer (Biolegend, San Diego, CA, USA) containing 5% Human TruStain FcX™ Fc Receptor Blocking Solution (BD Bioscience, San Jose, CA, USA) for 10 min. Cells were then probed with FITC conjugated anti-CD4 Multiclone SK3 and SK4 (Biolegend, San Diego, CA, USA), APC anti-CD8 clone SKI (Biolegend, San Diego, CA, USA), and V450 conjugated anti-CD19 clone SJ25C1 (BD Bioscience, San Jose, CA, USA) for 20 min at room temperature. Cells were washed and resuspended in PBS containing 2 µg/mL of PI and analyzed using a BD FACSVerse Flow Cytometer (BD Bioscience, San Jose, CA, USA). Data were collected using FACSuite™ software (BD Bioscience, San Jose, CA, USA) and analyzed using FlowJo software (BD Bioscience, San Jose, CA, USA).

### 4.9. Evalution of Pore Formation by Western Blot Analysis

PBMNC (1.5 × 10^6^ cells/mL) were incubated with activated ETX at 0 nM, 10 nM, 25 nM, and 50 nM for 120 min at 37 °C. Alternatively, cells were treated with 50 nM of ETX for 30, 60, or 120 min. As a positive control for pore formation, rMAL-CHO cells treated with or without 50 nM active ETX for 30 min were used as controls [7,78]. After treatment, cells were immediately moved to ice, then washed three times with ice cold PBS. Cells were lysed in ice-cold RIPA buffer (50 mM Tris-HCl (pH 8.0), 150 mM NaCl, +1% NP-40, 0.1% Sodium dodecyl sulfate, 0.5% Sodium Deoxycholate) with proteinase and phosphatase inhibitors (Cell Signaling Technologies, Danvers, MA, USA) for 10 min. Samples were centrifuged at 5000 rcf for 5 min to pellet nuclei and DNA. Supernatants were collected and used for Western blot analysis. All samples were prepared in 2X Laemmli Sample Buffer (Bio-Rad, Hercules, CA, USA) containing 5% 2-Mercaptoethanol (Bio-Rad, Hercules, CA, USA) and heated at 95 °C for 5 min before loading onto 4–20% Mini-PROTEAN TGX Stain-Free gels (Bio-Rad, Hercules, CA, USA). Gels were run in Tris/Glycine SDS Buffer (Bio-Rad, Hercules, CA, USA) at 200 V for 35 min. Semi-dry transfers were performed in transfer Tris/Glycine Buffer (Bio-Rad, Hercules, CA, USA), using the Trans-Blot SD Semi-Dry Electrophoretic Transfer Cell system (Bio-Rad, Hercules, CA, USA) at 15 V for 15 min. Blots were blocked in 5% Blotting-Grade Blocker nonfat milk (Bio-Rad) in Tris Buffered Saline with Tween 20 (TBS-T, Cell Signaling Technology, Danvers, MA, USA) for one hour at room temperature. Blots were then incubated with primary antibodies anti-ETX antibody JL004 at 0.34 µg/mL in blocking solution overnight at 4 °C. Blots were washed three times for 5 min in TBS-T at room temperature and incubated with secondary antibody peroxidase-conjugated Affinipure Goat Anti-Rabbit IgG H + L (Jackson ImmunoResearch, Baltimore, MD, USA) at 0.024 µg/mL in blocking solution for 2 h at room temperature. Blots were washed three times for 5 min in TBS-T and developed for 5 min at room temperature in SuperSignal West Dura Extended Duration Substrate (ThermoFisher Scientific, Waltham, MA, USA). The developed blots were visualized on 5 × 7 CL-XPosure Films (ThermoFisher Scientific, Waltham, MA, USA) at various exposure times using a Konica Minolta SRX-101A film processor.

### 4.10. Statistics

One-way ANOVA with post hoc Tukey HSD test was used to determine significance when comparing three or more data points. Unpaired Student’s t-tests were used to determine significance when comparing only two data points. These instances are indicated in figure legends.

## Figures and Tables

**Figure 1 toxins-15-00423-f001:**
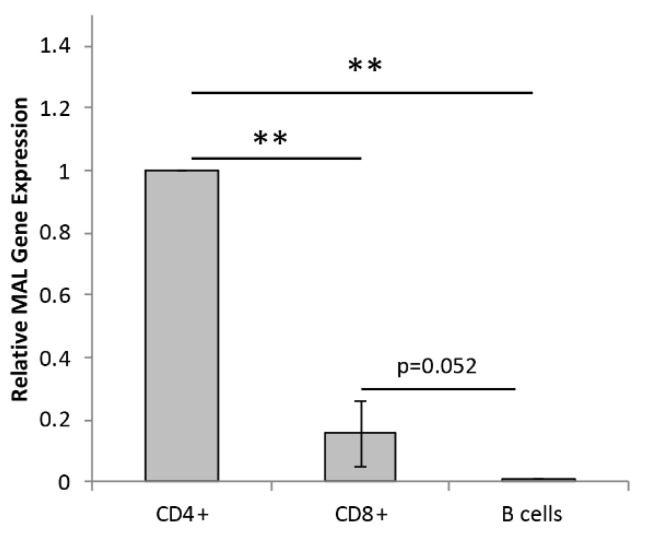
Real-time quantitative PCR (RT-qPCR) analysis of *Mal* gene expression in primary human lymphocytes. *Mal* transcripts levels were quantified in cDNAs obtained from isolated populations of CD4+, CD8+, and B cells. Relative *Mal* expression in isolated CD4+, CD8+, and B cells. B-actin was used as a reference gene. ** *p* < 0.01 determined by One-way ANOVA with post hoc Tukey HSD Test. Results are expressed as the mean performed in triplicate.

**Figure 2 toxins-15-00423-f002:**
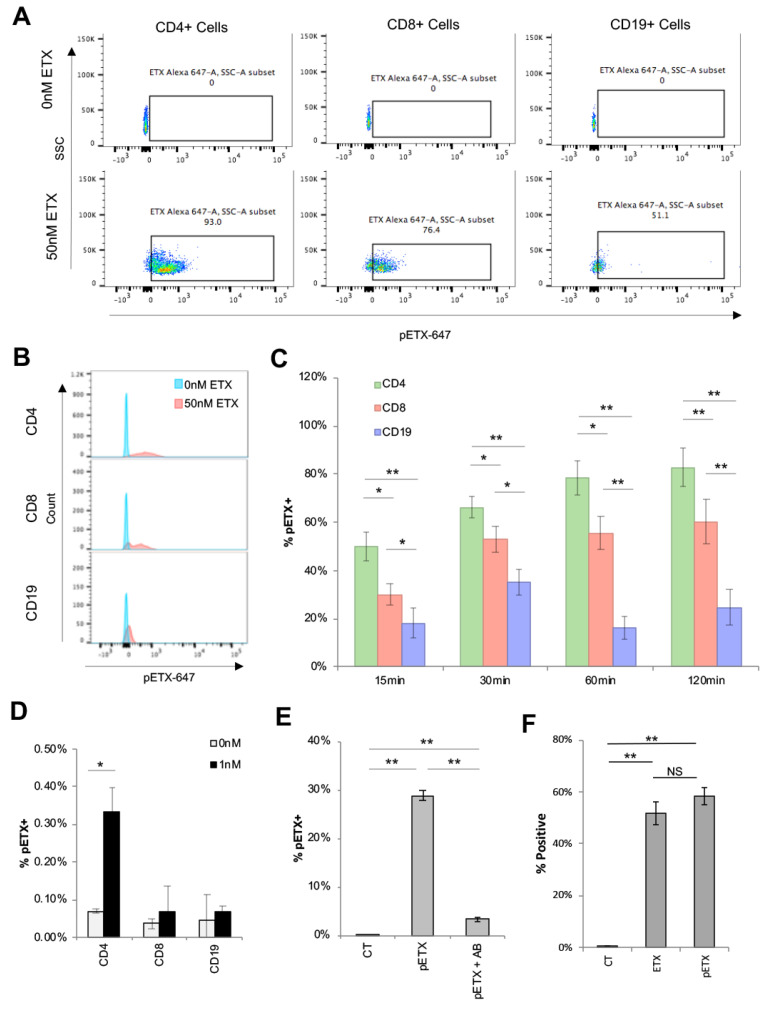
ETX binds to CD4+, CD8+, and CD19+ lymphocytes with a preference for CD4+ cells. PBMNCs were incubated with 0 nM or 50 nM pETX-647 for 2 h and binding to CD4+, CD8+, and CD19+ cells was examined by flow cytometry. An example of the gating strategy for examination of CD4+, CD8+, and CD19+ lymphocytes is depicted in Appendix A. Representative scatter plots (**A**) and histogram analysis (**B**) of pETX-647 fluorescent intensity from three separate donors performed in triplicate. (**C**) PBMNCs were incubated with 25 nM pETX-647 for indicated time points and binding determined by flow cytometry. Results are expressed as the percent of CD4+, CD8+, or CD19+ cells positive for pETX (% pETX+). Results are the means of three separate donors performed in triplicate. (**D**) PBMNCs were incubated with 0 nM or 1 nM pETX-647 for 2 h. Results are expressed as the percentages of CD4+, CD8+, or CD19+ cells positive for pETX (% pETX+). Results are the means of two separate donors performed in triplicate. (**E**) Anti-ETX antibody inhibits binding to CD4+ cells. Media containing 50 nM pETX-647 was pretreated with or without an antibody known to block ETX binding for 30 min before treating PBMNC for 2 h and evaluated by flow cytometry. Percent of pETX+ lymphocytes when cells are incubated without pETX-647 (CT), with pETX-647 (pETX), or pETX-647 pretreated with anti-ETX antibody (pETX + anti-ETX). (**F**) PBMNCs were treated with 25 nM of unlabeled pETX or ETX for 2 h. Untreated cells were used as controls. pETX and ETX binding to lymphocytes was determined using an affinity-purified anti-ETX polyclonal rabbit antibody and PE-conjugated anti-rabbit IgG. Results expressed as percent CD4+ lymphocytes positive for ETX or pETX (% Positive). Data points are the mean performed in triplicate. * *p* < 0.05, ** *p* < 0.01, determine by One-way ANOVA with post hoc Tukey HSD Test.

**Figure 3 toxins-15-00423-f003:**
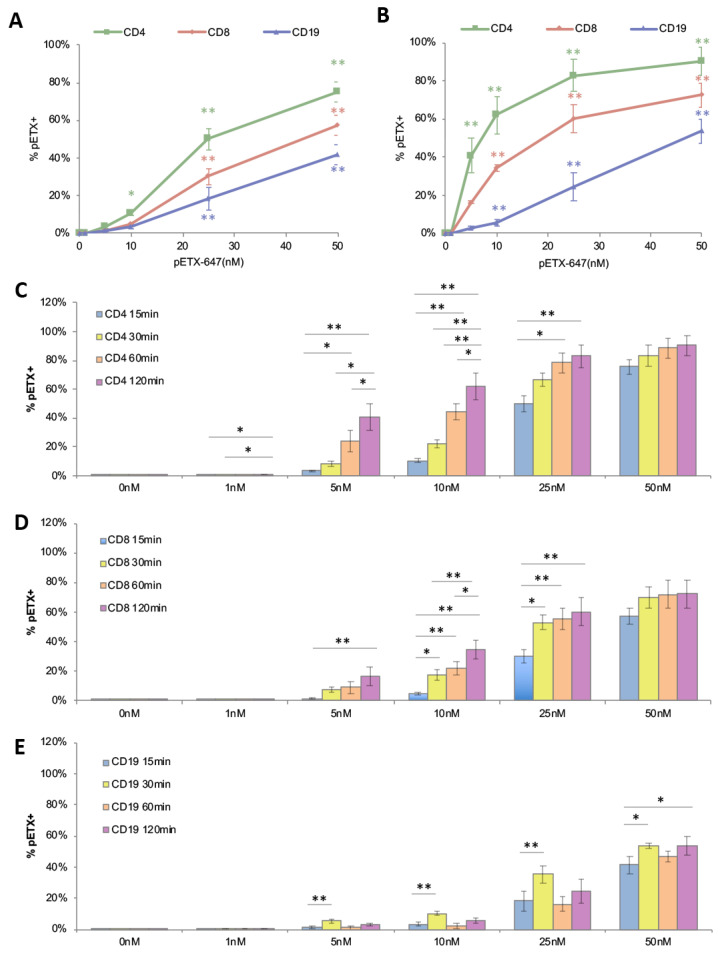
ETX bindings to lymphocytes are time and dose dependent. To determine if ETX binding is dose dependent, PBMNCs were incubated with indicated doses of pETX-647 for 15 (**A**) and 120 min (**B**), and pETX-647 binding was determined by flow cytometry. Results are expressed as percent pETX positive (% pETX+) cells for CD4+, CD8+, and CD19+ cells. * *p* < 0.05 and ** *p* < 0.001 compared to untreated controls (0 nM), as determined by One-way ANOVA with post hoc Tukey HSD Test. For a more detailed analysis of *p* values for all pETX doses, refer to Appendix A. To determine if ETX binding is time dependent, PBMNCs were incubated with 0 nM, 1 nM, 5 nM, 10 nM, 25 nM, or 50 nM pETX-647 for indicated time points (**C**–**E**). The percentages of ETX positive cells were determined by flow cytometry for CD4+ cells (**C**), CD8+ cells (**D**), and CD19+ cells (**E**). * *p* < 0.01 and ** *p* < 0.001 were determined by One-way ANOVA with post hoc Tukey HSD Test. All results are the mean results of three donors performed in triplicate.

**Figure 4 toxins-15-00423-f004:**
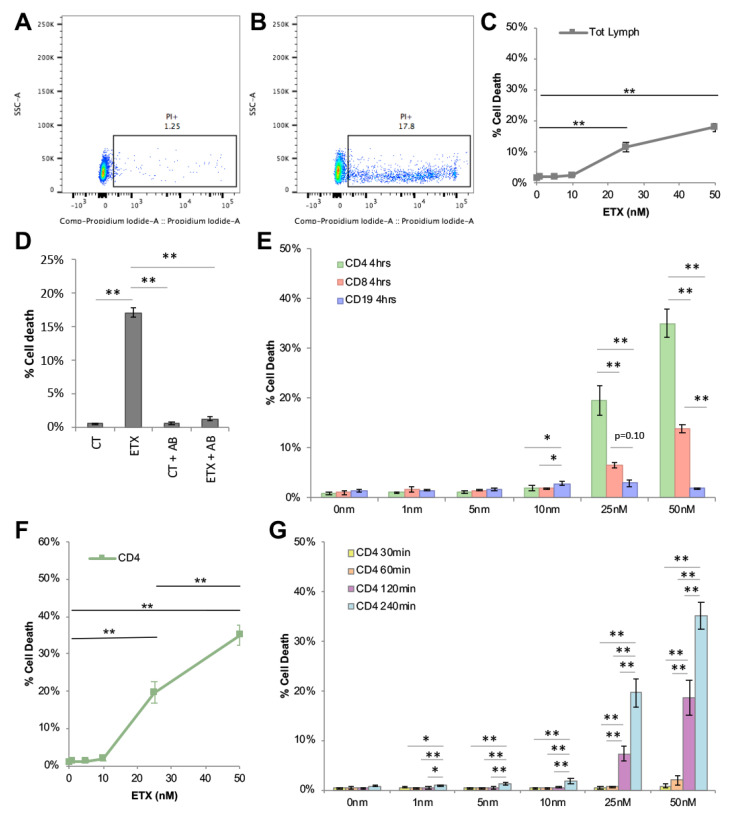
Active ETX kills human CD4+ cells in a dose- and time-dependent manner. PI inclusion was used to evaluate cell viability via flow cytometry. PI positive cells are considered dead. Results are expressed as the number of PI positive cells out of the total specific population and expressed as percent cell death (% cell death). (**A**,**B**) Representative scatter plots of total lymphocytes stained with PI to evaluate cell death after treatment with 0 nM (**A**) or 50 nM (**B**) ETX for 4 h. (**C**) Total lymphocyte cell death after 4 h of incubation with indicated ETX doses. (**D**) Media containing 50 nM ETX were pretreated with or without a neutralizing anti-ETX antibody for 30 min before treating lymphocytes for 2 h. Cell death was evaluated by PI inclusion via flow cytometry. Percent cell death when lymphocytes are treated in media alone (CT), media containing 50 nM ETX (ETX), media pretreated with anti-ETX antibody (CT + anti-ETX), and media containing 50 nM ETX pretreated with anti-ETX antibody (ETX + anti-ETX). (**E**) Percent cell death of different lymphocyte subsets at indicated ETX doses after 4 h of treatment. (**F**) Cell death was evaluated in CD4+ cells after 4 h of treatment at indicated ETX doses. These data are the same data depicted in Figure 4E with different statistical analyses. ETX-induced cell death of CD4+ cells is dose dependent. (**G**) ETX-induced cytotoxicity in CD4+ cells over time at various time points. Data are the mean of a single donor in triplicate. Data are representative of multiple donors. * *p* < 0.05, ** *p* < 0.01, determined by One-way ANOVA with post hoc Tukey HSD Test.

**Figure 5 toxins-15-00423-f005:**
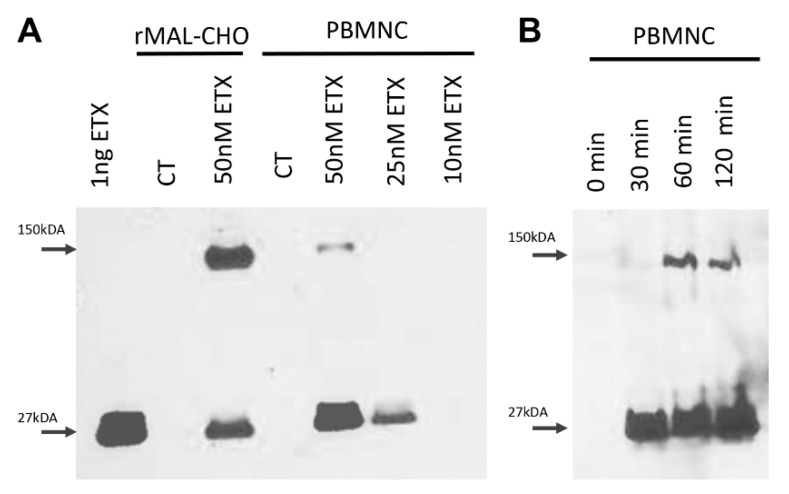
ETX cytotoxicity is mediated by pore formation in PBMNC. (**A**) PBMNCs were treated with indicated doses of ETX for 2 h. Cells were extensively washed in PBS, and whole-cell lysates were examined via Western blot for detection of the 150kDa oligomerized pore complex or 27 kDa bound ETX monomer. In total, 1 ng of ETX and whole-cell lysates from rMAL-CHO cells treated with or without 50 nM ETX were used as controls. (**B**) PBMNCs were treated with 50 nM of ETX for the indicated time points and examined for via Western blot for detection of the 150 kDa oligomerized pore complex or 27 kDa bound monomer.

## Data Availability

The data presented in this study are available in this article and supplementary materials.

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
