# Peer review of "Clostridium perfringens Epsilon Toxin Binds to and Kills Primary Human Lymphocytes"

_toxins, 2023, doi:10.3390/toxins15070423_

Round 1

Reviewer 1 Report

The present manuscript describes the effect of C. perfringens epsilon toxin on peripheral blood lymphocytes. The authors positively correlated MAL expression on CD4,8,19-positive lymphocytes with binding and cytotoxic effect of epsilon toxin. The authors also differentiated between monomers and pore forming oligomers of epsilon toxin that bound to PBMCs thereby explaining the difference between high number of cells that bound toxin and comparable low number of cell death. Overall, the manuscript is sound, but only summarizes results, that have been shown before. The novelty of the manuscript is the use of primary lymphocytes. The authors spent great effort to correlate MAL expression with binding and cytotoxic effect of epsilon toxin to verify MAL as receptor, which, however, still is under discussion. The study in its present form  shows one flaw that should be addressed in a further experiment:

Major point:

The authors performed binding studies of epsilon toxin over different time periods. That includes initial binding, oligomerization, and putative endosome recycling or MAL-dependent reconstruction of the plasma membrane as well as shuttle of MAL. These are highly dynamic processes. To correlate epsilon toxin binding with MAL surface exhibition the authors should perform binding studies at 4°C for limited time and only check initial binding. Since MAL is suggested to be a receptor for epsilon toxin, the abundance of cell surface protein and not total MAL protein should be correlated with binding of epsilon toxin. Commercially available MAL antibodies for flow cytometry that recognize extracellular loop 2 of MAL exist, that can be used for detection of surface MAL. Since ECL2 was assumed to interact with epsilon toxin, this antibody might by chance also interfere with toxin binding.

Minor points:

Please define “pathogenic” lymphocytes in line 40 and 398/399

Check legend figure 2 d,e,f

Author Response

We thank the reviewer for their insight and time.  Our response are italicized and highlighted.  

The present manuscript describes the effect of C. perfringens epsilon toxin on peripheral blood lymphocytes. The authors positively correlated MAL expression on CD4,8,19-positive lymphocytes with binding and cytotoxic effect of epsilon toxin. The authors also differentiated between monomers and pore forming oligomers of epsilon toxin that bound to PBMCs thereby explaining the difference between high number of cells that bound toxin and comparable low number of cell death. Overall, the manuscript is sound, but only summarizes results, that have been shown before. The novelty of the manuscript is the use of primary lymphocytes. The authors spent great effort to correlate MAL expression with binding and cytotoxic effect of epsilon toxin to verify MAL as receptor, which, however, still is under discussion. The study in its present form shows one flaw that should be addressed in a further experiment:

Major point:

The authors performed binding studies of epsilon toxin over different time periods. That includes initial binding, oligomerization, and putative endosome recycling or MAL-dependent reconstruction of the plasma membrane as well as shuttle of MAL. These are highly dynamic processes. To correlate epsilon toxin binding with MAL surface exhibition the authors should perform binding studies at 4°C for limited time and only check initial binding.

We agree, that is why cell binding experiments were performed with epsilon protoxin (pETX) which does not oligomerize or form pores, preventing endosome recycling and possible cell surface rearrangement of MAL1-14.  We have emphasized this in the text.

Since MAL is suggested to be a receptor for epsilon toxin, the abundance of cell surface protein and not total MAL protein should be correlated with binding of epsilon toxin. Commercially available MAL antibodies for flow cytometry that recognize extracellular loop 2 of MAL exist, that can be used for detection of surface MAL. Since ECL2 was assumed to interact with epsilon toxin, this antibody might by chance also interfere with toxin binding.

We agree with the reviewer that evaluation of MAL protein expression is preferred over RNA expression and would prefer this analysis.  However, we have purchased and tested numerous commercially available anti-MAL antibodies as well as privately generated antibodies with very little success.  The only assay with minimally acceptable results has been a western blot (WB) assay with Santa Cruz’s anti-MAL antibody sc-390687.  However, to detect MAL expression via WB, we require the maximal 40ug of lymphocyte cell lysate per well of our electrophoresis gels (4–20% Mini-PROTEAN® TGX™ Precast Protein Gels, 10-well, 50 µl #4561094) to reliably detect a band.  Because we are trying to compare MAL expression levels of CD4+, CD8+, and CD19+ cells isolated from the same donor, the amount of protein from a single donor to check all three cell types is a total of 120ug.  This is prohibitive, as the volume of blood required to meet the minimum protein requirement for each cell type from a single donor exceeds what our blood donors are willing to donate.  Based on previous experiments using our isolation methods described in the paper, we typically isolate 2 x10^5 CD4+ cells, 1x10^5 cells, and 1x10^4 CD19 cells from 1mL of peripheral blood.   We typically obtain 0.5ug of protein from a total of 1x10^4 lymphocytes.  Therefore we need 8x10^5 lymphocytes per cell type to achieve 40ug per well.  Therefore, to obtain the minimal required amount of protein for each cell type, we need 4mL of total blood for CD4+ cells, 8mls of whole blood for CD8+ cells is, and 80mls of whole blood for CD19+ cells, or a total  of 92mL of whole blood from a single donor (or ~20 vacutainer vials).  This far exceeds the normal volume of blood a typical donor is willing to donate (~40mL or 10 vacutainers) and exceeds the maximal volume of blood approved on our Human Subjects IRB protocol (50mL).  Even if we were to isolate our cells using FACS, we still need a minimum of 80mL of total blood from each donor.  Taken together, these limitations make completing this experiment incredibly difficult.  To address this limitation, we have expanded the discussion to include a section stating the weakness of examining RNA versus protein expression. 

Minor points:

Please define “pathogenic” lymphocytes in line 40 and 398/399.

We have better defined this in the text.

Check legend figure 2 d,e,f

Thank you, we have amended this. 

  1. Robertson, S. L.; Li, J.;  Uzal, F. A.; McClane, B. A., Evidence for a prepore stage in the action of Clostridium perfringens epsilon toxin. PLoS One 2011, 6 (7), e22053.
  2. Nagahama, M.; Ochi, S.; Sakurai, J., Assembly of Clostridium perfringens epsilon-toxin on MDCK cell membrane. J Nat Toxins 1998, 7 (3), 291-302.
  3. Petit, L.; Gibert, M.;  Gillet, D.;  Laurent-Winter, C.;  Boquet, P.; Popoff, M. R., Clostridium perfringens epsilon-toxin acts on MDCK cells by forming a large membrane complex. J Bacteriol 1997, 179 (20), 6480-7.
  4. Dorca-Arevalo, J.; Gomez de Aranda, I.; Blasi, J., New Mutants of Epsilon Toxin from Clostridium perfringens with an Altered Receptor-Binding Site and Cell-Type Specificity. Toxins (Basel) 2022, 14 (4).
  5. Miyata, S.; Matsushita, O.;  Minami, J.;  Katayama, S.;  Shimamoto, S.; Okabe, A., Cleavage of a C-terminal peptide is essential for heptamerization of Clostridium perfringens epsilon-toxin in the synaptosomal membrane. J Biol Chem 2001, 276 (17), 13778-83.
  6. Miyata, S.; Minami, J.;  Tamai, E.;  Matsushita, O.;  Shimamoto, S.; Okabe, A., Clostridium perfringens epsilon-toxin forms a heptameric pore within the detergent-insoluble microdomains of Madin-Darby canine kidney cells and rat synaptosomes. J Biol Chem 2002, 277 (42), 39463-8.
  7. Rumah, K. R.; Ma, Y.;  Linden, J. R.;  Oo, M. L.;  Anrather, J.;  Schaeren-Wiemers, N.;  Alonso, M. A.;  Fischetti, V. A.;  McClain, M. S.; Vartanian, T., The Myelin and Lymphocyte Protein MAL Is Required for Binding and Activity of Clostridium perfringens epsilon-Toxin. PLoS Pathog 2015, 11 (5), e1004896.
  8. Linden, J. R.; Flores, C.;  Schmidt, E. F.;  Uzal, F. A.;  Michel, A. O.;  Valenzuela, M.;  Dobrow, S.; Vartanian, T., Clostridium perfringens epsilon toxin induces blood brain barrier permeability via caveolae-dependent transcytosis and requires expression of MAL. PLoS Pathog 2019, 15 (11), e1008014.
  9. Anton, O. M.; Andres-Delgado, L.;  Reglero-Real, N.;  Batista, A.; Alonso, M. A., MAL protein controls protein sorting at the supramolecular activation cluster of human T lymphocytes. J Immunol 2011, 186 (11), 6345-56.
  10. Llorente, A.; de Marco, M. C.; Alonso, M. A., Caveolin-1 and MAL are located on prostasomes secreted by the prostate cancer PC-3 cell line. J Cell Sci 2004, 117 (Pt 22), 5343-51.
  11. Millan, J.; Puertollano, R.;  Fan, L.;  Rancano, C.; Alonso, M. A., The MAL proteolipid is a component of the detergent-insoluble membrane subdomains of human T-lymphocytes. Biochem J 1997, 321 ( Pt 1), 247-52.
  12. Nagahama, M.; Itohayashi, Y.;  Hara, H.;  Higashihara, M.;  Fukatani, Y.;  Takagishi, T.;  Oda, M.;  Kobayashi, K.;  Nakagawa, I.; Sakurai, J., Cellular vacuolation induced by Clostridium perfringens epsilon-toxin. FEBS J 2011, 278 (18), 3395-407.
  13. Ventimiglia, L. N.; Alonso, M. A., Biogenesis and Function of T Cell-Derived Exosomes. Front Cell Dev Biol 2016, 4, 84.
  14. Ventimiglia, L. N.; Fernandez-Martin, L.;  Martinez-Alonso, E.;  Anton, O. M.;  Guerra, M.;  Martinez-Menarguez, J. A.;  Andres, G.; Alonso, M. A., Cutting Edge: Regulation of Exosome Secretion by the Integral MAL Protein in T Cells. J Immunol 2015, 195 (3), 810-4.
  15. Blanch, M.; Dorca-Arevalo, J.;  Not, A.;  Cases, M.;  Gomez de Aranda, I.;  Martinez-Yelamos, A.;  Martinez-Yelamos, S.;  Solsona, C.; Blasi, J., The Cytotoxicity of Epsilon Toxin from Clostridium perfringens on Lymphocytes Is Mediated by MAL Protein Expression. Mol Cell Biol 2018, 38 (19).
  16. Kohno, T.; Moriuchi, R.;  Katamine, S.;  Yamada, Y.;  Tomonaga, M.; Matsuyama, T., Identification of genes associated with the progression of adult T cell leukemia (ATL). Jpn J Cancer Res 2000, 91 (11), 1103-10.
  17. Lara-Lemus, R., On The Role of Myelin and Lymphocyte Protein (MAL) In Cancer: A Puzzle With Two Faces. J Cancer 2019, 10 (10), 2312-2318.
  18. Rubio-Ramos, A.; Labat-de-Hoz, L.;  Correas, I.; Alonso, M. A., The MAL Protein, an Integral Component of Specialized Membranes, in Normal Cells and Cancer. Cells 2021, 10 (5).
  19. Chan, J. K., Mediastinal large B-cell lymphoma: new evidence in support of its distinctive identity. Adv Anat Pathol 2000, 7 (4), 201-9.
  20. Copie-Bergman, C.; Gaulard, P.;  Maouche-Chretien, L.;  Briere, J.;  Haioun, C.;  Alonso, M. A.;  Romeo, P. H.; Leroy, K., The MAL gene is expressed in primary mediastinal large B-cell lymphoma. Blood 1999, 94 (10), 3567-75.
  21. Copie-Bergman, C.; Plonquet, A.;  Alonso, M. A.;  Boulland, M. L.;  Marquet, J.;  Divine, M.;  Moller, P.;  Leroy, K.; Gaulard, P., MAL expression in lymphoid cells: further evidence for MAL as a distinct molecular marker of primary mediastinal large B-cell lymphomas. Mod Pathol 2002, 15 (11), 1172-80.
  22. Hsi, E. D.; Sup, S. J.;  Alemany, C.;  Tso, E.;  Skacel, M.;  Elson, P.;  Alonso, M. A.; Pohlman, B., MAL is expressed in a subset of Hodgkin lymphoma and identifies a population of patients with poor prognosis. Am J Clin Pathol 2006, 125 (5), 776-82.
  23. Ma, Y.; Sannino, D.;  Linden, J. R.;  Haigh, S.;  Zhao, B.;  Grigg, J. B.;  Zumbo, P.;  Dundar, F.;  Butler, D.;  Profaci, C. P.;  Telesford, K.;  Winokur, P. N.;  Rumah, K. R.;  Gauthier, S. A.;  Fischetti, V. A.;  McClane, B. A.;  Uzal, F. A.;  Zexter, L.;  Mazzucco, M.;  Rudick, R.;  Danko, D.;  Balmuth, E.;  Nealon, N.;  Perumal, J.;  Kaunzner, U.;  Brito, I. L.;  Chen, Z.;  Xiang, J. Z.;  Betel, D.;  Daneman, R.;  Sonnenberg, G. F.;  Mason, C. E.; Vartanian, T., Epsilon toxin-producing Clostridium perfringens colonize the multiple sclerosis gut microbiome overcoming CNS immune privilege. J Clin Invest 2023, 133 (9).

Author Response

We thank the reviewer for their time and insight.  We have italicized and highlighted our responses.

Clostridium perfringens epsilon toxin (ETX) is known to target the central nervous system in livestock and a potential trigger of multiple sclerosis in humans. Previous studies identified MAL, a protein expressed by myelin forming cells and specific epithelia in mammals and in T cells in humans, as the receptor for ETX. This manuscript shows that human CD4+ T cells have higher levels of MAL mRNA than CD8+ T cells and CD19+ B cells, and that the extent of ETX binding and killing in these three cell types correlates with the relative abundance of MAL transcripts. It was already known that ETX binds to and kills human T cells (Blanch et al., Mol Cell Biol 2018, 38, 200086-18; Shorkrzadeh et al. Microb Pathogen 2021, 156, 104820), and that CD4+ T cells express more MAL protein than CD8+ T cells (Copie-Bergman et al., 2002. Mod Pathol 15, I172-I180). The findings that CD19+ B cells have low levels of MAL transcripts and that ETX binds to CD19+ peripheral B cells, despite the reported lack of MAL protein expression in B cells, are unexpected. Since cell purification never reaches 100% efficiency, the results regarding this aspect of the study are unclear, as the enriched CD19+ cell sample likely contains residual T cells that could account for the observed MAL transcripts, the binding of ETX, and cell killing. In conclusion, the part of the study on CD19+ cells and ETX is not robust, and the rest of the paper does not offer substantial new information.

We agree that Mal expression in B-cells and ETX binding to CD19+ cells was unexpected.  Because we use negative selection to isolate B-cells for our gene expression analysis, we agree that our B-cell populations use for the RT-PCR experiments  may not be a 100% pure and have addressed this limitation in the revised manuscript.  To address this limitation,  we have compared our results to previously published datasets examining MAL expression in different lymphocyte population, summarized in Supplementary figure 1.   These datasets used several different methods to both isolate and/or identify CD19+/B-cells.  Isolation methods included FACS sorting (Supp Fig 1A and B), positive magnetic selection (Supp Fig 1C), and single cell RNAseq analysis (Supp Fig D-F).  Mal expression was evaluated using RNAseq (Supp Fig 1A, B, and D) and microarray (Supp Fig 1C).  These results also demonstrated a low but still detectable levels of MAL transcripts in CD19/B-cells, consistent with our RT-PCR results.  Based on these finding, we believe our RT-PCR results are accurate. We have added text elaborating on the weakness of our B-cell isolation method and the findings of these other datasets.  

As far as ETX binding to CD19+ cells are concerned, this was determined using multicolor flow cytometry.  PBMNC were simultaneously probed with FITC conjugated anti-CD4, PE conjugated anti-CD8, and V450 conjugated anti-CD19 antibody as well as Alexa647 conjugate pETX (pETX-647). Therefore, only cells that expressed CD4, CD8, or CD19 individually were evaluated for pETX binding as demonstrated in Supplementary figure 2.  Therefore, 100% of CD19 cells express CD19.  The methodology and gating strategy have been emphasized in the text and figure 2 legend. 

Reviewer 3 Report

The manuscript presented by the Authors contains a thorough study to show the effects of epsilon toxin produced by Clostridium perfringens strains on human cells and the immune response in neurological diseases resulting from ETX binding to human lymphocytes. I like the care and detail with which the Authors described the various stages of the study and the insight with which the results obtained were interpreted. The manuscript contains significant elements of novelty testifying to its high substantive value.

Author Response

We thank the reviewer for their time and insight.  We have italicized and highlighted our responses. 

The manuscript presented by the Authors contains a thorough study to show the effects of epsilon toxin produced by Clostridium perfringens strains on human cells and the immune response in neurological diseases resulting from ETX binding to human lymphocytes. I like the care and detail with which the Authors described the various stages of the study and the insight with which the results obtained were interpreted. The manuscript contains significant elements of novelty testifying to its high substantive value.

We thank the viewer for their feedback. 

Reviewer 4 Report

This paper confirms in primary cells what Blanch et al (2018: reference in list is incomplete) already discovered in cell lines. However, in the present study the direct link to MAL as a receptor (binding and blocking assays)  for epsilon toxin is not shown, so expanding on this in the discussion and even suggesting this in the title is misleading. Moreover, the word correlation should only be used when it is statistically documented, so the word correlates should certainly not be in the title of this paper. The research gap and the purpose of the study is not accurately delineated. I could not find the important reference 3 in the literature. Lines 447 and 448: I don't think it is phosphatase but phosphate. It is not clear where the blood samples for isolating PBMNC originated from.

Author Response

We thank the reviewers for their time and insight.  We have italicized and highlighted our responses. 

This paper confirms in primary cells what Blanch et al (2018: reference in list is incomplete) already discovered in cell lines. However, in the present study the direct link to MAL as a receptor (binding and blocking assays)  for epsilon toxin is not shown, so expanding on this in the discussion and even suggesting this in the title is misleading. Moreover, the word correlation should only be used when it is statistically documented, so the word correlates should certainly not be in the title of this paper. The research gap and the purpose of the study is not accurately delineated. I could not find the important reference 3 in the literature.

We agree and have removed “correlation” from the title and text.  We have replaced “correlation” “positive association” where appropriate in the text.

We agree that our findings confirm the findings Blanch et al15 observed in lymphocyte cell lines but respectfully disagree that our findings are not of importance.  Blanch et al used Jurkat, MOLT-4 ,TK6 and JeKo-1 lymphocyte cell lines derived from an acute T cell leukemia patient (cells derived from blood), an acute lymphoblastic leukemia patient, an  hereditary spherocytosis  patient (cells derived from spleen),and a mantle cell lymphoma patient (cells derived from blood), respectively.  Importantly, the ETX sensitive Jurkat and MOLT-4 cells are cancerous, lymphoblast cell lines.    Because MAL dysregulation has been repeatedly reported in cancer and cancerous cell lines including lymphoblastic leukemic and B-cell lymphomas16-22, it’s imperative that ETX binding and sensitivity be confirmed on primary lymphocyte cells, especially those isolated from healthy donors. The importance of this confirmation has been emphasized in the text. 

Reference 3 (Ma, Y.;  Sannino, D.;  Linden, J. R.;  Haigh, S.;  Zhao, B.;  Grigg, J. B.;  Zumbo, P.;  Dundar, F.;  Butler, D. J.;  Profaci, C. P.;  Telesford, K. M.;  Winokur, P. N.;  Rumah, K. R.;  Gauthier, S. A.;  Fischetti, V. A.;  McClane, B. A.;  Uzal, F. A.;  Zexter, L.;  Mazzucco, M.;  Rudick, R.;  Danko, D.;  Balmuth, E.;  Nealon, N.;  Perumal, J.;  Kaunzner, U. W.;  Brito, I. L.;  Chen, Z.;  Xiang, J. Z.;  Betel, D.;  Daneman, R.;  Sonnenberg, G. F.;  Mason, C. E.; Vartanian, T., Epsilon toxin-producing Clostridium perfringens colonize the MS gut and epsilon toxin overcomes immune privilege. J Clin Invest 2023.) was availble as a preprint on the JCI website at the first submission of this manuscript.  It has been finalzied and the title changed to “Epsilon toxin–producing Clostridium perfringens colonize the multiple sclerosis gut microbiome overcoming CNS immune privilege” and is now publically avalable at JCI website (https://www.jci.org/articles/view/163239)23.  The citation has been updated.    

Lines 447 and 448: I don't think it is phosphatase but phosphate.

Thank you, we have corrected this.

It is not clear where the blood samples for isolating PBMNC originated from. 

We are unclear if the reviewer is asking about the methodology used to isolate the PBMNC or the demographic information of PBMNC donors.  The original manuscript has a detailed methodology section outlining where and how PBMNC were isolated.  A brief summary of the method has been added to the results section. If the reviewer is referring to donor demographics, we have included a short section summarizing demographics in the methods section.

  1. Robertson, S. L.; Li, J.;  Uzal, F. A.; McClane, B. A., Evidence for a prepore stage in the action of Clostridium perfringens epsilon toxin. PLoS One 2011, 6 (7), e22053.
  2. Nagahama, M.; Ochi, S.; Sakurai, J., Assembly of Clostridium perfringens epsilon-toxin on MDCK cell membrane. J Nat Toxins 1998, 7 (3), 291-302.
  3. Petit, L.; Gibert, M.;  Gillet, D.;  Laurent-Winter, C.;  Boquet, P.; Popoff, M. R., Clostridium perfringens epsilon-toxin acts on MDCK cells by forming a large membrane complex. J Bacteriol 1997, 179 (20), 6480-7.
  4. Dorca-Arevalo, J.; Gomez de Aranda, I.; Blasi, J., New Mutants of Epsilon Toxin from Clostridium perfringens with an Altered Receptor-Binding Site and Cell-Type Specificity. Toxins (Basel) 2022, 14 (4).
  5. Miyata, S.; Matsushita, O.;  Minami, J.;  Katayama, S.;  Shimamoto, S.; Okabe, A., Cleavage of a C-terminal peptide is essential for heptamerization of Clostridium perfringens epsilon-toxin in the synaptosomal membrane. J Biol Chem 2001, 276 (17), 13778-83.
  6. Miyata, S.; Minami, J.;  Tamai, E.;  Matsushita, O.;  Shimamoto, S.; Okabe, A., Clostridium perfringens epsilon-toxin forms a heptameric pore within the detergent-insoluble microdomains of Madin-Darby canine kidney cells and rat synaptosomes. J Biol Chem 2002, 277 (42), 39463-8.
  7. Rumah, K. R.; Ma, Y.;  Linden, J. R.;  Oo, M. L.;  Anrather, J.;  Schaeren-Wiemers, N.;  Alonso, M. A.;  Fischetti, V. A.;  McClain, M. S.; Vartanian, T., The Myelin and Lymphocyte Protein MAL Is Required for Binding and Activity of Clostridium perfringens epsilon-Toxin. PLoS Pathog 2015, 11 (5), e1004896.
  8. Linden, J. R.; Flores, C.;  Schmidt, E. F.;  Uzal, F. A.;  Michel, A. O.;  Valenzuela, M.;  Dobrow, S.; Vartanian, T., Clostridium perfringens epsilon toxin induces blood brain barrier permeability via caveolae-dependent transcytosis and requires expression of MAL. PLoS Pathog 2019, 15 (11), e1008014.
  9. Anton, O. M.; Andres-Delgado, L.;  Reglero-Real, N.;  Batista, A.; Alonso, M. A., MAL protein controls protein sorting at the supramolecular activation cluster of human T lymphocytes. J Immunol 2011, 186 (11), 6345-56.
  10. Llorente, A.; de Marco, M. C.; Alonso, M. A., Caveolin-1 and MAL are located on prostasomes secreted by the prostate cancer PC-3 cell line. J Cell Sci 2004, 117 (Pt 22), 5343-51.
  11. Millan, J.; Puertollano, R.;  Fan, L.;  Rancano, C.; Alonso, M. A., The MAL proteolipid is a component of the detergent-insoluble membrane subdomains of human T-lymphocytes. Biochem J 1997, 321 ( Pt 1), 247-52.
  12. Nagahama, M.; Itohayashi, Y.;  Hara, H.;  Higashihara, M.;  Fukatani, Y.;  Takagishi, T.;  Oda, M.;  Kobayashi, K.;  Nakagawa, I.; Sakurai, J., Cellular vacuolation induced by Clostridium perfringens epsilon-toxin. FEBS J 2011, 278 (18), 3395-407.
  13. Ventimiglia, L. N.; Alonso, M. A., Biogenesis and Function of T Cell-Derived Exosomes. Front Cell Dev Biol 2016, 4, 84.
  14. Ventimiglia, L. N.; Fernandez-Martin, L.;  Martinez-Alonso, E.;  Anton, O. M.;  Guerra, M.;  Martinez-Menarguez, J. A.;  Andres, G.; Alonso, M. A., Cutting Edge: Regulation of Exosome Secretion by the Integral MAL Protein in T Cells. J Immunol 2015, 195 (3), 810-4.
  15. Blanch, M.; Dorca-Arevalo, J.;  Not, A.;  Cases, M.;  Gomez de Aranda, I.;  Martinez-Yelamos, A.;  Martinez-Yelamos, S.;  Solsona, C.; Blasi, J., The Cytotoxicity of Epsilon Toxin from Clostridium perfringens on Lymphocytes Is Mediated by MAL Protein Expression. Mol Cell Biol 2018, 38 (19).
  16. Kohno, T.; Moriuchi, R.;  Katamine, S.;  Yamada, Y.;  Tomonaga, M.; Matsuyama, T., Identification of genes associated with the progression of adult T cell leukemia (ATL). Jpn J Cancer Res 2000, 91 (11), 1103-10.
  17. Lara-Lemus, R., On The Role of Myelin and Lymphocyte Protein (MAL) In Cancer: A Puzzle With Two Faces. J Cancer 2019, 10 (10), 2312-2318.
  18. Rubio-Ramos, A.; Labat-de-Hoz, L.;  Correas, I.; Alonso, M. A., The MAL Protein, an Integral Component of Specialized Membranes, in Normal Cells and Cancer. Cells 2021, 10 (5).
  19. Chan, J. K., Mediastinal large B-cell lymphoma: new evidence in support of its distinctive identity. Adv Anat Pathol 2000, 7 (4), 201-9.
  20. Copie-Bergman, C.; Gaulard, P.;  Maouche-Chretien, L.;  Briere, J.;  Haioun, C.;  Alonso, M. A.;  Romeo, P. H.; Leroy, K., The MAL gene is expressed in primary mediastinal large B-cell lymphoma. Blood 1999, 94 (10), 3567-75.
  21. Copie-Bergman, C.; Plonquet, A.;  Alonso, M. A.;  Boulland, M. L.;  Marquet, J.;  Divine, M.;  Moller, P.;  Leroy, K.; Gaulard, P., MAL expression in lymphoid cells: further evidence for MAL as a distinct molecular marker of primary mediastinal large B-cell lymphomas. Mod Pathol 2002, 15 (11), 1172-80.
  22. Hsi, E. D.; Sup, S. J.;  Alemany, C.;  Tso, E.;  Skacel, M.;  Elson, P.;  Alonso, M. A.; Pohlman, B., MAL is expressed in a subset of Hodgkin lymphoma and identifies a population of patients with poor prognosis. Am J Clin Pathol 2006, 125 (5), 776-82.
  23. Ma, Y.; Sannino, D.;  Linden, J. R.;  Haigh, S.;  Zhao, B.;  Grigg, J. B.;  Zumbo, P.;  Dundar, F.;  Butler, D.;  Profaci, C. P.;  Telesford, K.;  Winokur, P. N.;  Rumah, K. R.;  Gauthier, S. A.;  Fischetti, V. A.;  McClane, B. A.;  Uzal, F. A.;  Zexter, L.;  Mazzucco, M.;  Rudick, R.;  Danko, D.;  Balmuth, E.;  Nealon, N.;  Perumal, J.;  Kaunzner, U.;  Brito, I. L.;  Chen, Z.;  Xiang, J. Z.;  Betel, D.;  Daneman, R.;  Sonnenberg, G. F.;  Mason, C. E.; Vartanian, T., Epsilon toxin-producing Clostridium perfringens colonize the multiple sclerosis gut microbiome overcoming CNS immune privilege. J Clin Invest 2023, 133 (9).

Round 2

Reviewer 1 Report

All points are adequately addressed.

Reviewer 2 Report

My concerns have been adequately addressed.

Reviewer 4 Report

I have no further comments